# Impact of Bariatric Surgery on Subtilisin/Kexin Type 9 (PCSK9) Gene Expression and Inflammation in the Adipose Tissue of Obese Diabetic Rats

**DOI:** 10.3390/ijms242316978

**Published:** 2023-11-30

**Authors:** Adrian H. Heald, Helene A. Fachim, Bilal Bashir, Bethanie Garside, Safwaan Adam, Zohaib Iqbal, Akheel A. Syed, Rachelle Donn, Carel W. Le Roux, Mahmoud Abdelaal, James White, Handrean Soran

**Affiliations:** 1Faculty of Biology, Medicine and Health, University of Manchester, Oxford Road, Manchester M13 9PT, UK; helene.fachim@manchester.ac.uk (H.A.F.); bilal.bashir@doctors.org.uk (B.B.); bethanie.garside@nhs.net (B.G.); safwaan.adam@manchester.ac.uk (S.A.); zahaib@doctors.org.uk (Z.I.); akheel.syed@nca.nhs.uk (A.A.S.); rachelle.donn@manchester.ac.uk (R.D.); hsoran@aol.com (H.S.); 2Department of Diabetes, Endocrinology and Obesity Medicine, Salford Royal NHS Foundation Trust, Stott Lane, Salford M6 8HD, UK; 3Cardiovascular Trials Unit, Manchester University NHS Foundation Trust, Oxford Road, Manchester M13 9WL, UK; 4The Christie Hospital NHS Foundation Trust, Wilmslow Road, Manchester M20 4BX, UK; 5Diabetes Complications Research Centre, Conway Institute, School of Medicine, University College Dublin, D04 V1W8 Dublin, Ireland; carel.leroux@ucd.ie (C.W.L.R.); mahmoud.abdelaal@ucd.ie (M.A.); james.white@ucd.ie (J.W.)

**Keywords:** bariatric surgery, Zucker Diabetic Sprague Dawley, diabetes, obesity, PCSK9, fat tissue

## Abstract

Bariatric surgery improves dyslipidaemia and reduces body weight, but it remains unclear how bariatric surgery modulates gene expression in fat cells to influence the *proprotein convertase subtilisin/kexin type 9 (PCSK-9)* and *low-density lipoprotein receptor (LDLR)* gene expression. The expression of the *PCSK9/LDLR/tumor necrosis factor-alpha (TNFα)* gene in adipose tissue was measured in two groups of Zucker Diabetic Sprague Dawley (ZDSD) rats after Roux-en-Y gastric bypass (RYGB) surgery or ‘SHAM’ operation. There was lower *PCSK9* (*p* = 0.02) and higher *LDLR* gene expression (*p* = 0.02) in adipose tissue in rats after RYGB. Weight change did not correlate with *PCSK9* gene expression (r = −0.5, *p* = 0.08) or *TNFα* gene expression (r = −0.4, *p* = 0.1). *TNFα* gene expression was positively correlated with *PCSK9* gene expression (r = 0.7, *p* = 0.001) but not correlated with *LDLR* expression (r = −0.3, *p* = 0.3). Circulating triglyceride levels were lower in RYGB compared to the SHAM group (1.1 (0.8–1.4) vs. 1.5 (1.0–4.2), *p* = 0.038) mmol/L with no difference in cholesterol levels. *LDLR* gene expression was increased post-bariatric surgery with the potential to reduce the number of circulating LDL particles. *PCSK9* gene expression and *TNFα* gene expression were positively correlated after RYGB in ZDSD rats, suggesting that the modulation of pro-inflammatory pathways in adipose tissue after RYGB may partly relate to *PCSK9* and *LDLR* gene expression.

## 1. Introduction

Obesity is linked to a notable reduction in life expectancy and has been connected to over 5 million global fatalities in 2019. The highest prevalence of overweight and obesity is observed in European regions, reaching an estimated 59%, resulting in more than 1.2 million annual fatalities. The impact of obesity and its associated health issues has considerable implications for healthcare services. In the United States, where obesity prevalence is 41.9%, the yearly healthcare expenses related to obesity amounted to USD 173 billion in 2019 [1].

Obesity is considered a chronic inflammatory state driven by increased adiposity where adipose tissue and its host cells, predominantly macrophages, act like a toxic endocrine organ, produce various cytokines and hormones, causing an imbalance between pro-inflammatory and anti-inflammatory cytokines, which are central to the pathogenesis of the complications of obesity [2,3].

Bariatric surgery improves metabolic comorbidities that are commonly associated with obesity. These improvements encompass dyslipidemia, reduced blood pressure, remission of type 2 diabetes, decreased insulin resistance, and diminished systemic inflammation. Moreover, the surgery exerts a decelerating effect on the progression of atherosclerosis, thereby contributing to a reduction in both cardiovascular-specific risks and overall mortality rates [4].

The function of adipocytes in processing low-density lipoprotein cholesterol (LDL-C) and the impact of proprotein covertase subtilisin/kexin type 9 (PCSK9) on adipocyte low-density lipoprotein receptor (LDLR) regulation are both poorly understood. Important investigations on LDLR and adipocytes were conducted more than 40 years ago. A high-affinity receptor that can bind, internalise, and degrade LDL was found in isolated human adipose cells by Angel et al. in 1979, indicating that adipose tissue is a significant location of LDL and high-density lipoprotein (HDL) interactions [5]. Since then, the function of adipose tissue in the metabolism of lipoproteins has either been mostly ignored or has not received much attention.

Marked changes in lipid metabolism are also seen post bariatric surgery in humans including increase in HDL cholesterol (HDL-C), enhance HDL quality, reductions in triglycerides, LDL-C, insulin resistance, systemic and adipose tissue inflammation and improve LDL quality [3,6]. PCSK9 controls the number of LDLRs. These receptors play a critical role in regulating blood cholesterol levels through the binding and removal of circulating LDL particles, the primary carriers of cholesterol in the blood. LDLRs are particularly abundant in the liver but are also present in fat tissue. The number of LDLRs on the surface of cells determines how quickly LDL particles and their cholesterol cargo are removed from the bloodstream [7]. When PCSK9 attaches to LDLRs, it leads to the breakdown of LDLRs before they are recycled to the cell surface, resulting in more cholesterol remaining in the bloodstream.

Zucker Diabetic Sprague Dawley (ZDSD) rats are selectively inbred for obesity, insulin resistance and diabetes [8]. As the ZDSD rat has an intact leptin pathway, the metabolic origin of the type 2 diabetes (T2D)-like state that spontaneously develops in the ZDSD rat strain is a more translatable model to the clinical metabolic syndrome in humans [9]. The data on how bariatric surgery modulates gene expression in fat cells are still limited. In this study, we examined how bariatric surgery may modulate *PCSK9* and *LDLR* gene expression in ZDSD rats.

## 2. Results

RYGB surgery reduced the bodyweight of rats as expected (Figure 1, Table 1). The gene expression results for *PCSK9*, *LDLR*, and *TNFα* are shown in Figure 2. There was significantly lower *PCSK9* gene expression in adipose tissue in rats after RYGB (*p* = 0.03). Correspondingly, adipose tissue *LDLR* gene expression was higher in the RYGB rats (*p* = 0.03). We observed a trend of lower *TNFα* gene expression (*p* = 0.07) after RYGB (Figure 2). TNFα protein levels did not significantly differ between RYGB- and SHAM-operated rats (*p* = 0.11) (Table 1).

We observed a positive correlation between the *TNFα* and *PCSK9* gene expression (r = 0.7, *p* = 0.001 (Figure 3)) but no correlation between *TNFα* and *LDLR* gene expression (r = −0.3, *p* = 0.3). Weight change did not correlate with *PCSK9*, *TNFα* and *LDLR* gene expression (r = −0.5, *p* = 0.08; r = −0.4, *p* = 0.1 and r = 0.08, *p* = 0.8, respectively). (Appendix A). Higher *PCSK9* gene expression was not correlated with TNFα protein in either group.

In Figure 4, we have shown the specific *PSCK9* gene expression and *TNFα* gene expression (fold changes) for the RYGB- and SHAM-treated rats separately.

Circulating triglyceride [median (min-max)] levels were lower in RYGB compared to the SHAM group (1.1 (0.8–1.4) vs. 1.5 (1.0–4.2), *p* = 0.038) but cholesterol was comparable in RYGB compared to the SHAM group [median (min-max) 2.2 (1.7–2.4) vs. 2.7 (2.0–3.1), *p* = 0.11) (Table 1).

## 3. Discussion

This study investigates the impact of Roux-en-Y gastric bypass surgery on gene expression in adipose tissue, specifically focusing on the *PCSK9* and *LDLR* genes. This research is noteworthy due to its unique focus on the effects of bariatric surgery on these specific genes and their potential implications for lipid metabolism and cardiovascular health. In this study conducted using ZDSD rats, we showed that *LDLR* gene expression was increased post bariatric surgery with the potential to reduce the number of circulating LDL particles. *PCSK9* gene expression was lower in RYGB rats and was positively correlated with *TNFα* gene expression, suggesting that the modulation of pro-inflammatory pathways in adipose tissue after RYGB may partly relate to *PCSK9* and *LDLR* gene expression. The changes in the *PCSK9* gene and *LDLR* gene expression were not associated with the degree of weight loss.

The impact of RYGB on *PCSK9*, *LDLR* and *TNFα* gene expression, compared to a sham procedure, suggests a possible mechanism of RYGB on adipose tissue inflammation, with a subsequent impact on lipid pathways. While the adipose tissue expression of *PCSK9* and *LDLR* makes less of a contribution to circulating cholesterol levels than expression in the liver, their contribution to ameliorate the pro-inflammatory environment from adipose tissue may be important [7].

Exogenously administrated PCSK9 reduces LDLR protein levels in the liver, while the action of PCSK9 on the LDLR is not dependent on catalytic activity in vivo [10]. PCSK9 expression is regulated by the transcription factor insulin/AKT2 and hepatocyte nuclear factor 1 alpha (HNF1α). Additionally, increased mammalian ‘target of rapamycin complex 1′ (mTORC1) activity leads to the activation of protein kinase C delta (PKCδ), with reduced activity of hepatocyte nuclear factor 4 alpha (HNF4α) and HNF1α, decreased PCSK9 expression, and ultimately increased hepatic LDLR protein levels, with resultant decreased circulating LDL particles and cholesterol levels [11]. As a result, it is probable that various inputs to mTORC1 contribute to the substantial activation of LDLR in obese mice with its potential benefits in relation to a less atherogenic lipid profile.

Adipose dysfunction caused by “inflammation”, as seen in insulin-resistance conditions, may impair HDL lipidation in the adipocytes, reducing circulating HDL-C levels [12]. In a clinical study, we demonstrated that RYGB enhances HDL functionality and ABCA1 gene expression in gluteal subcutaneous adipose tissue in association with a reduction in adipose tissue and systemic inflammation [13]. However, in this study, there were no significant adipose tissue ABCA1 gene expression between the RYGB and SHAM groups. This is likely related to the small sample size.

In humans, adipose tissue plays a role in lipoprotein metabolism. Notably, when the expression of PCSK9 is induced, LDLR is reduced through PCSK9-mediated degradation. On the contrary, when the expression of PCSK9 is reduced due to insulin, LDLR degradation is also reduced [14]. The preliminary results that we report here suggest that adipose tissues may be an important milieu in relation to how RYGB influences the regulation of these processes and also influence *TNFα* gene expression.

This novel study delves into the impact of RYGB surgery on the expression of crucial genes, *PCSK9* and *LDLR*, in adipose tissue, shedding light on their role in cholesterol regulation. The observed increase in *LDLR* gene expression post-surgery suggests a potential mechanism for lowering LDL particle count. Furthermore, the intriguing correlation between *PCSK9* and *TNF*α gene expression hints at a link between inflammatory pathways and PCSK9 modulation. This complex interplay adds depth to our understanding of inflammation, adipose tissue, and lipid metabolism, with implications for emerging drug therapies targeting PCSK9. Ultimately, these insights may pave the way for innovative interventions targeting *PCSK9* and *LDLR* gene expression in obesity and metabolic disorders, offering promising avenues to enhance lipid profiles and mitigate cardiovascular risk in obese individuals. We have not reported circulating LDL cholesterol levels in the rats. We accept this as a limitation.

A strength of this paper is that the experiment was conducted under highly regulated conditions in relation to calorie intake. A limitation is that we did not have access to liver tissue and hence could not confirm our findings in the liver. In the future, we plan to extend the work to other relevant candidate genes with a larger sample size in terms of the numbers of rats studied.

## 4. Materials and Methods

### 4.1. Animal Model—Approved by the Board of UCD Conway

The protocol was prepared and registered with The Board of UCD Conway’s Ethical Review Committee, who approved the use of animals in this study.

Obese male ZDSD rats were obtained and divided into two groups: SHAM (Sham laparotomy-operated obese diabetic rats; n = 8) and RYGB (Obese diabetic rats undergoing RYGB surgery; n = 9).

Measurement of body weight was carried out at intervals. 

### 4.2. Gene and Protein Expression

In this study, the expression of the PCSK9, LDLR and tumor necrosis factor alpha *(TNFα)* gene in adipose tissue in 2 groups of ZDSD rats was determined in relation to intervention with RYGB surgery or a ‘SHAM’ operation. RNA was extracted from gluteal adipose tissue and reverse transcribed. The cDNA concentration and purity were determined using a NanoDrop Lite spectrophotometer (ThermoFisher Scientific, Cambridge, Cambridgeshire, UK). Relative gene expression was determined using a StepOne Plus (ThermoFisher Scientific) following the steps: Pre-incubation—1 cycle at 95 °C for 10 min, Amplification—50 cycles at 95ºC for 10 s, 60 °C for 30 s and 72 °C for 1 s, Cooling—1 cycle, 40 °C for 30 s. The assays were conducted using Real-Time quantitative PCR (qPCR) using β-actin as a housekeeping gene. We also measured TNF α protein in fat tissue homogenates performed via ELISA. A Rat TNF-α Quantikine ELISA Kit (R&D Systems, Abingdon, UK, Cat no RTA00) consisting of a Solid Phase Sandwich ELISA was used. We used 0.5 mg of tissue in 1× PBS and lysis buffer (1:3, total 1 mL) and used a homogenizator to break up the tissue after overnight incubation at 30 °C. Then, the samples were centrifuged at maximum speed, 10 min at 4 °C; the supernatant was collected and used in our assays. Nanodrop was used to determine the protein concentration. The samples were diluted 1000× before starting the ELISA assay. Reagents and kits were obtained from Applied Biosystems (ThermoFisher Scientific, Cambridge, Cambridgeshire, UK) (TaqMan assays) for gene expression and reverse transcription and R&D for ELISA.

### 4.3. Theory/Calculations

Gene expression was quantified using the Comparative threshold (Ct) method (ΔΔCt Method, ΔΔCT = ΔCT (a target sample) – ΔCT (a reference sample)) [15,16], and the amount of target gene was normalized to the housekeeping gene *ACTB* and determined by 2^−ΔΔCt^, with relative expression levels reported as fold change (RYGB normalized against SHAM group) [median (min-max)]. Data normality was determined using the Shapiro–Wilk Normality test and via the direct visualisation of the histogram and normal Q-Q plots. We performed comparisons between the control group SHAM and RYGB using independent sample Mann–Whitney U test for non-parametric data. Statistical analysis was performed using the “Statistical Package for Social Sciences” (SPSS) version 22.0 (IBM Corp: Armonk, NY, USA) and GraphPad Prism v 9.3.1 (471). Values of *p* ≤ 0.05 were considered significant. We tested for correlations of PCSK9 gene expression fold change with LDLR and TNF alpha gene expression and change in weight, using Spearman correlation. 

## 5. Conclusions

We speculate that RYGB surgery in ZDSD rats might lead to a reduction in *PCSK9* gene expression and a potential increase in *LDLR* gene expression, which could potentially result in a decrease in circulating LDL particle count. The observed shifts in *TNFα* gene expression hint at a speculative connection between the modification of pro-inflammatory pathways in adipose tissue following RYGB and alterations in *PCSK9* and *LDLR* expression.

## Figures and Tables

**Figure 1 ijms-24-16978-f001:**
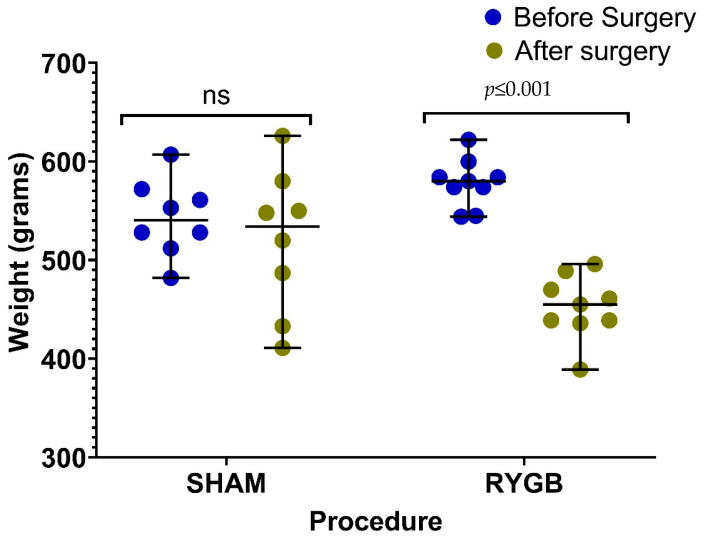
Comparison of pre-operative and post-operative body weights in subjects undergoing both the SHAM (n = 8) and RYGB (n = 9). Significant changes in body mass following RYGB but not with SHAM procedure. Results were compared using Mann–Whitney U test and shown as median (min-max). ns: non-significative.

**Figure 2 ijms-24-16978-f002:**
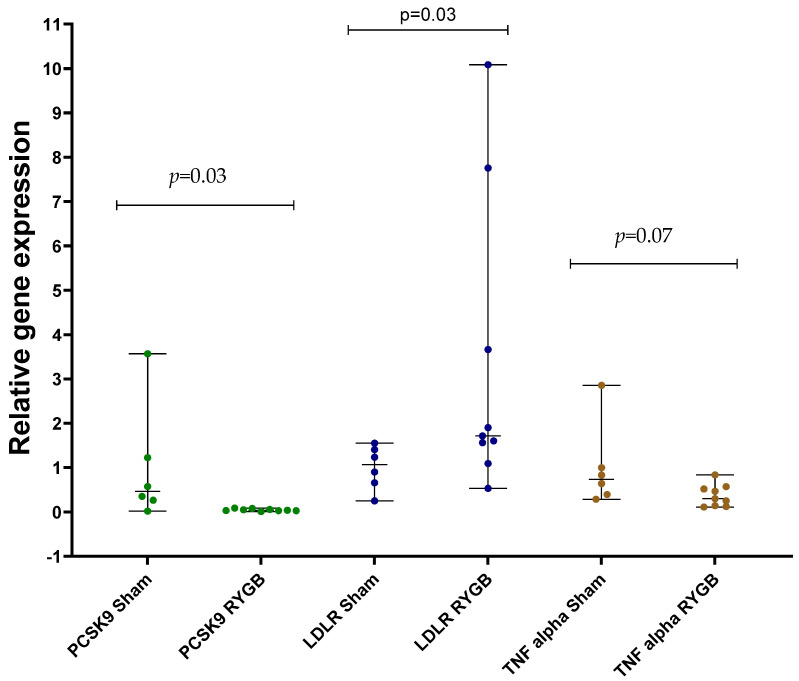
Comparative analysis of gene expression profiles in adipose tissue examining the expression levels of *LDLR*, *PCSK9*, and *TNFα*, in rats that have undergone RYGB (n = 8) surgery versus SHAM (n = 7) surgical procedure. Data were obtained by 2^ΔΔCt^ calculation using ACTB as housekeeping gene and fold change normalized against SHAM group. Results were analyzed using Mann–Whitney U test and shown as median (min-max). LDLR: Low-density lipoprotein receptor, PCSK9: proprotein covertase subtilisin/kexin type 9, RYGB: Roux-en-Y Gastric bypass, TNFα: Tumor necrosis factor alpha.

**Figure 3 ijms-24-16978-f003:**
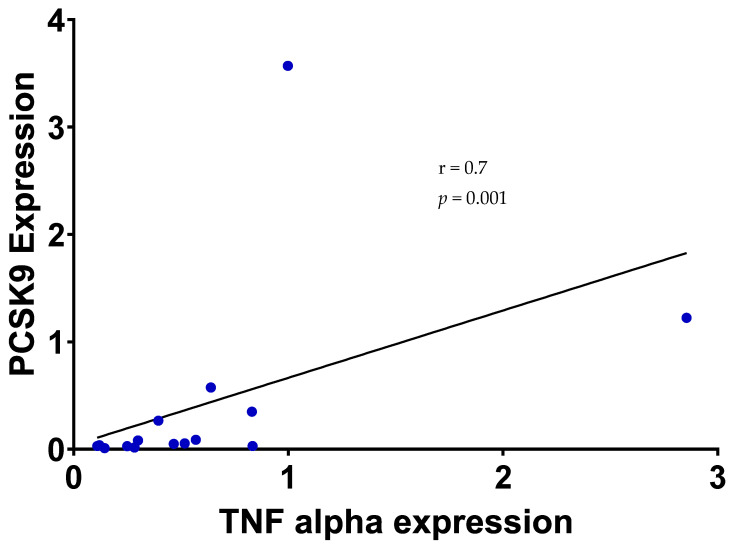
Positive Spearman’s correlation between *PCSK9* and *TNFα* gene expression in adipose tissue in SHAM and RYGB rats.

**Figure 4 ijms-24-16978-f004:**
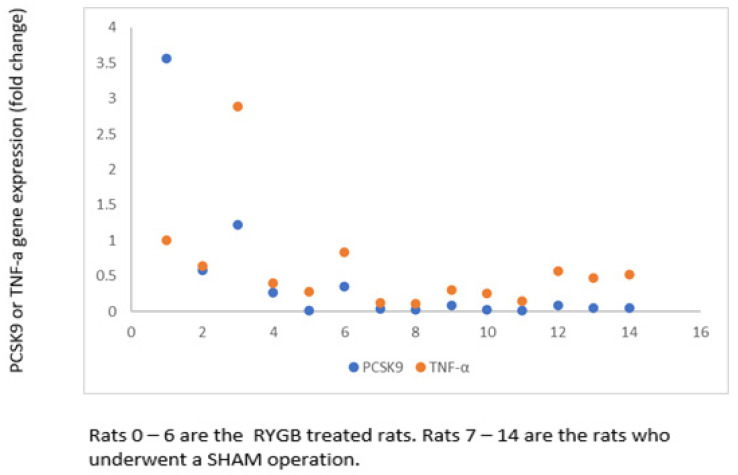
*TNFα* and *PSCK9* fold change for RYGB and SHAM-treated rats.

**Table 1 ijms-24-16978-t001:** Comparison of change in weight, gene expression, concentration of TNFα and triglyceride between RYGB and SHAM procedure.

Variable	RYGB	SHAM	*p* Value
% Change in weight (gms)	−20.26 (16.26–32.23)	+1.28 * (−9.4–22.26)	0.01
PCSK9 **	0.04 (0.01–0.09)	0.46 (0.02–3.57)	0.02
LDLR **	1.71 (0.53–10.09)	1.07 (0.25–1.56)	0.02
TNFα Protein (pg/mL)	24.36 (19.71–25.61)	25.34 (22.93–25.61)	0.11
TG mmol/L ^§^	1.1 (0.8–1.4) ^§^	1.5 (1.0–4.2)	0.04

* +denotes gain in weight. ** expressed as fold change. ^§^ median (range).

## Data Availability

This study data will be available from the corresponding author upon reasonable request.

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
