# Peer review of "Impact of Bariatric Surgery on Subtilisin/Kexin Type 9 (PCSK9) Gene Expression and Inflammation in the Adipose Tissue of Obese Diabetic Rats"

_ijms, 2023, doi:10.3390/ijms242316978_

Round 1

Reviewer 1 Report

Comments and Suggestions for Authors

Dear authors, I have reviewed your interesting manuscript entitled: ‘ABC Subtilisin/Kexin Type 9 (PCSK9) Gene Expression and Inflammation In Adipose Tissue In Obese Diabetic Rats’. The manuscript is clearly written, methodically correct and the results are adequately discussed. The text could be enriched with this information:

Major points:
1. Authors should strongly emphasize the unique character of the paper, why this study is important?, why their work is valuable in the field?
2. How this findings can be applicated to clinical trials?
3. What is a practical implication of the study?
4. In the future, I recommend the authors to expand their work with other experiments that would focus on (1) larger sample size, and (2) other genes. These plans could be briefly introduced in Discussion.
5. The experiment was performed in limited series thus I suggest to modify conclusions into more speculative ones.
6. You are extremely short on the ELISA test I would suggest 2-3 more sentences, including catalog number.
 7. Please provide  detailed fat tissue homogenization procedure.

Minor point:
1. In line 37 and 38 abbreviations should be explained.

Author Response

Reviewer 1:

Dear authors, I have reviewed your interesting manuscript entitled: ‘ABC Subtilisin/Kexin Type 9 (PCSK9) Gene Expression and Inflammation In Adipose Tissue In Obese Diabetic Rats’. The manuscript is clearly written, methodically correct and the results are adequately discussed. The text could be enriched with this information:

Major points:
1. Authors should strongly emphasize the unique character of the paper, why this study is important?, why their work is valuable in the field?

Response: Following sentence added in opening paragraph of discussion.

The study presented investigates the impact of Roux-en-Y gastric bypass surgery on gene expression in adipose tissue, specifically focusing on the PCSK9 and LDLR genes. This research is particularly noteworthy due to its unique focus on the effects of bariatric surgery on these specific genes and their potential implications for lipid metabolism and cardiovascular health”.

And following towards the end of discussion.

This novel study delves into the impact of RYGB surgery on the expression of crucial genes, PCSK9 and LDLR, in adipose tissue, shedding light on their role in cholesterol regulation. The observed increase in LDLR gene expression post-surgery suggests a potential mechanism for lowering LDL-particle count. Furthermore, the intriguing correlation between PCSK9 and TNFα gene expression hints at a link between inflammatory pathways and PCSK9 modulation. This complex interplay adds depth to our understanding of inflammation, adipose tissue, and lipid metabolism, with implications for emerging drug therapies targeting PCSK9. Ultimately, these insights may pave the way for innovative interventions targeting PCSK9 and LDLR gene expression in obesity and metabolic disorders, offering promising avenues to enhance lipid profiles and mitigate cardiovascular risk in obese individuals.

  1. How this findings can be applicated to clinical trials?

The positive correlation between PCSK9 and TNFα gene expression after bariatric surgery suggests a potential link between inflammatory pathways and PCSK9 regulation. Clinical trials could explore interventions that target inflammatory pathways to modulate PCSK9 expression. Clinical trials could assess whether the observed changes in gene expression after bariatric surgery is replicable in humans and are consistent across different patient populations. This information could be used to develop personalized treatment approaches that target specific gene expression profiles.

  1. What is a practical implication of the study?

If the observed changes in gene expression are replicated in humans and the findings are consistent across different populations, the information could be used to develop personalized treatment approaches that target specific gene expression profiles.

  1. In the future, I recommend the authors to expand their work with other experiments that would focus on (1) larger sample size, and (2) other genes. These plans could be briefly introduced in Discussion.

Thank you for valuable suggestion.

  1. The experiment was performed in limited series thus I suggest to modify conclusions into more speculative ones.

Thanks for this suggestion. We have rephrased the conclusion.

  1. You are extremely short on the ELISA test I would suggest 2-3 more sentences, including catalog number.

Thanks for suggestion. We have included this detail in section 2.2

  1. Please provide  detailed fat tissue homogenization procedure.
    We have included this detail in section 2.2

Minor point:
1. In line 37 and 38 abbreviations should be explained.

Thanks for comments. We have expanded the abbreviations.

Reviewer 2 Report

Comments and Suggestions for Authors

The manuscript prepared by Heald et al. presents the results of interesting study aimed to investigate how bariatric surgery may modulate expression of genes i.e. PCSK9,  LDLR, TNF-α in adipose tissue of ZDSD rats.  

Major comments:

1.       In the introduction section, please add the following information:

·         epidemiological data on obesity;

·         what are the consequences of obesity?

·         how does bariatric surgery reduce cardiovascular morrtality?

2.       In the materials and methods section (gene and proteijn expression); please add where the reagents/kits for RNA isolation, reverse transcription and ELISAs come from.

3.       How many times were the prepared homogenates/samples for the ELISA assay diluted?

4.       Are you sure the gene expression data is expressed as 2-ΔΔCt and not as 2-ΔCt ?

5.       Was the compliance of the data distribution with the normal distribution checked before applying  Mann-Whitney U test?  If yes, please add information that the distribution of data was not in accordance with the normal distribution and the Mann-Whitney U test was applied.

6.       The descriptions under the figures are too short.

7.       There is no title and description above table 1. Please add.

8.       Please add a figure showing the TNF-α protein level results.

9.       Please add a table showing the results of the correlation (R, p).

10.   Why the 3rd study group (control group without surgical interventions) was not included in the study?

Author Response

Reviewer 2:

The manuscript prepared by Heald et al. presents the results of interesting study aimed to investigate how bariatric surgery may modulate expression of genes i.e. PCSK9,  LDLR, TNF-α in adipose tissue of ZDSD rats.  

Major comments:

  1. In the introduction section, please add the following information:
  • epidemiological data on obesity;

Thanks for suggestion. Added in opening paragraph of introduction.

  • what are the consequences of obesity?

A short paragraph added in introduction.

  • how does bariatric surgery reduce cardiovascular mortality?

A short paragraph added in introduction.

  1. In the materials and methods section (gene and protein expression); please add where the reagents/kits for RNA isolation, reverse transcription and ELISAs come from.

Thanks. Explanation has been added in section 2.2.

  1. How many times were the prepared homogenates/samples for the ELISA assay diluted?

1000x – added in the section 2.2.

  1. Are you sure the gene expression data is expressed as 2-ΔΔCtand not as 2-ΔCt ?

It is expressed as 2-ΔΔCt . (https://pubmed.ncbi.nlm.nih.gov/11846609/)

  1. Was the compliance of the data distribution with the normal distribution checked before applying  Mann-Whitney U test?  If yes, please add information that the distribution of data was not in accordance with the normal distribution and the Mann-Whitney U test was applied.

Yes, Data was not normally distributed and hence non parametric test was employed. Relevant sentence added in section

  1. The descriptions under the figures are too short.

Descriptions has been expanded.

  1. There is no title and description above table 1. Please add.

It has been provided now.

  1. Please add a figure showing the TNF-α protein level results.

TNF alpha protein concentration has been summarized in Table 1. Adding a figure will be duplication of a finding whose results are non-significant.

  1. 9.       Please add a table showing the results of the correlation (R, p).

Response: we are not sure what the reviewer wants here. In any event all relevant corelations have been recorded in the text

  1. Why the 3rd study group (control group without surgical interventions) was not included in the study?

Response

There was no group without any surgical intervention. The group who had SHAM procedure acted as a control group.

Round 2

Reviewer 1 Report

Comments and Suggestions for Authors

The authors implemented all my suggestions, therefore I endorse publication of the manuscript in its current form.

Author Response

Many thanks for your suggestions and support in our manuscript. 

Reviewer 2 Report

Comments and Suggestions for Authors

I am satisfied with the corrections introduced by the authors. But one point still requires clarification from the authors.

Please write step by step how gene expression was calculated.

How ∆ Ct was calculated and then how ∆∆ Ct was calculated?

e.g. for the PCSK9 gene:

1. ∆Ct RYGB group = Ct PCSK9 - Ct ACTB

 ∆Ct SHAM group = Ct PCSK9 - Ct ACTB

2. ∆∆ Ct = ∆ Ct RYGB group - ∆ Ct SHAM group

Please explain step by step how gene expression was calculated.

Author Response

Many thanks for your comments, suggestions and support. 

The  2^DDCt calculation was done based on 

ΔΔCT = ΔCT(a target sample)−ΔCT(a reference sample) = (CTD − CTB)−(CTC − CTA).

 ∆Ct RYGB group = Ct PCSK9 - Ct ACTB

∆Ct SHAM group = Ct PCSK9 - Ct ACTB

 ∆∆ Ct = ∆ Ct RYGB group - ∆ Ct SHAM group

Please see attachment of new manuscript with this information added.

Round 3

Reviewer 2 Report

Comments and Suggestions for Authors

Accept in present form.

Author Response

Thank you very much!